# Differential Infiltration of Key Immune T-Cell Populations Across Malignancies Varying by Immunogenic Potential and the Likelihood of Response to Immunotherapy

**DOI:** 10.3390/cells13231993

**Published:** 2024-12-03

**Authors:** Islam Eljilany, Sam Coleman, Aik Choon Tan, Martin D. McCarter, John Carpten, Howard Colman, Abdul Rafeh Naqash, Igor Puzanov, Susanne M. Arnold, Michelle L. Churchman, Daniel Spakowicz, Bodour Salhia, Julian Marin, Shridar Ganesan, Aakrosh Ratan, Craig Shriver, Patrick Hwu, William S. Dalton, George J. Weiner, Jose R. Conejo-Garcia, Paulo Rodriguez, Ahmad A. Tarhini

**Affiliations:** 1Departments of Cutaneous Oncology, H. Lee Moffitt Cancer Center and Research Institute, Tampa, FL 33612, USA; 2Huntsman Cancer Institute, Salt Lake City, UT 84132, USA; 3University of Colorado Cancer Center, Aurora, CO 80045, USA; 4Norris Comprehensive Cancer Center, University of Southern California, Los Angeles, CA 90033, USA; 5Department of Neurosurgery, School of Medicine, University of Utah, Salt Lake City, UT 84132, USA; 6Oklahoma University Health Stephenson Cancer Center, Oklahoma City, OK 73104, USA; 7Department of Medicine, Roswell Park Comprehensive Cancer Center, Buffalo, NY 14263, USA; 8University of Kentucky Markey Cancer Center, Lexington, KY 40536, USA; 9Aster Insights, Hudson, FL 34667, USA; 10Ohio State University Comprehensive Cancer Center, Columbus, OH 43210, USA; 11Simon Comprehensive Cancer Center, Indiana University, Indianapolis, IN 46202, USA; 12Rutgers Cancer Institute of New Jersey, New Brunswick, NJ 08903, USA; 13Department of Genome Sciences, School of Medicine, University of Virginia, Charlottesville, VA 22908, USA; 14Murtha Cancer Center, Walter Reed National Military Medical Center, Falls Church, VA 22042-5101, USA; 15Department of Internal Medicine, Carver College of Medicine, University of Iowa Health Care, Iowa City, IA 52242, USA; 16Department of Immunology, H. Lee Moffitt Cancer Center and Research Institute, Tampa, FL 33612, USA

**Keywords:** immune cell infiltration, immune response, melanoma, RNA expression, ovarian cancer, pancreatic cancer, bladder cancer, T-cell

## Abstract

**Background:** Solid tumors vary by the immunogenic potential of the tumor microenvironment (TME) and the likelihood of response to immunotherapy. The emerging literature has identified key immune cell populations that significantly impact immune activation or suppression within the TME. This study investigated candidate T-cell populations and their differential infiltration within different tumor types as estimated from mRNA co-expression levels of the corresponding cellular markers. **Methods:** We analyzed the mRNA co-expression levels of cellular biomarkers that define stem-like tumor-infiltrating lymphocytes (TILs), tissue-resident memory T-cells (TRM), early dysfunctional T-cells, late dysfunctional T-cells, activated-potentially anti-tumor (APA) T-cells and Butyrophilin 3A (BTN3A) isoforms, utilizing clinical and transcriptomic data from 1892 patients diagnosed with melanoma, bladder, ovarian, or pancreatic carcinomas. Real-world data were collected under the Total Cancer Care Protocol and the Avatar^®^ project (NCT03977402) across 18 cancer centers. Furthermore, we compared the survival outcomes following immune checkpoint inhibitors (ICIs) based on immune cell gene expression. **Results:** In melanoma and bladder cancer, the estimated infiltration of APA T-cells differed significantly (*p* = 4.67 × 10^−12^ and *p* = 5.80 × 10^−12^, respectively) compared to ovarian and pancreatic cancers. Ovarian cancer had lower TRM T-cell infiltration than melanoma, bladder, and pancreatic (*p* = 2.23 × 10^−8^, 3.86 × 10^−28^, and 7.85 × 10^−9^, respectively). Similar trends were noted with stem-like, early, and late dysfunctional T-cells. Melanoma and ovarian expressed BTN3A isoforms more than other malignancies. Higher densities of stem-like TILs; TRM, early and late dysfunctional T-cells; APA T-cells; and BTN3A isoforms were associated with increased survival in melanoma (*p* = 0.0075, 0.00059, 0.013, 0.005, 0.0016, and 0.041, respectively). The TRM gene signature was a moderate predictor of survival in the melanoma cohort (AUROC = 0.65), with similar findings in testing independent public datasets of ICI-treated patients with melanoma (AUROC 0.61–0.64). **Conclusions:** Key cellular elements related to immune activation are more heavily infiltrated within ICI-responsive versus non-responsive malignancies, supporting a central role in anti-tumor immunity. In melanoma patients treated with ICIs, higher densities of stem-like TILs, TRM T-cells, early dysfunctional T-cells, late dysfunctional T-cells, APA T-cells, and BTN3A isoforms were associated with improved survival.

## 1. Introduction

In 2023, melanoma, ovarian, pancreatic, and bladder cancers were projected to be among the most diagnosed cancers in the United States. The survival rates over five years for these cancers show considerable variation, with notably better outcomes observed for cutaneous melanoma and bladder carcinoma. This discrepancy in survival rates is partially attributed to the effectiveness of immunotherapy treatments in managing these cancers [1]. Immune checkpoint inhibitors (ICIs) have emerged as highly effective treatments for various advanced cancers, delivering extended survival benefits [2]. Specifically, therapies targeting immune checkpoints such as cytotoxic T-lymphocyte antigen-4 (CTLA-4) and programmed death/ligand 1 (PD-1/PD-L1) have demonstrated significant benefits in treating melanoma and bladder carcinoma [2].

However, many patients fail to respond favorably to these therapies, with some experiencing early disease progression due to mechanisms that confer tumor immune resistance [2]. Furthermore, ICI activity is relatively limited in treating some cancer types like ovarian cancer and pancreatic adenocarcinoma [3]. Notably, tumor immune resistance can be primarily attributed to the distinctive features of their tumor microenvironments (TMEs), which play pivotal roles in cancer development and immune resistance mechanisms [4]. The immunogenic nature of the TME impacts tumor immune evasion through various complex mechanisms [5].

The TME represents a sophisticated ecosystem incorporating diverse cell types that interact within their architectures. Immune cells infiltrating tumors are pivotal in modulating tumor behavior and susceptibility to therapeutic interventions [6,7]. Notably, tumor antigen-specific CD8+ cytotoxic T-cells play a crucial function in anti-tumor immunity, detecting and eliminating tumor cells that express neoantigens and distinctive antigens arising from mutated gene expressions [8]. However, the tumor milieu also includes immunosuppressive cells, including regulatory T (Treg)-cells and myeloid-derived suppressor cells (MDSCs), which can foster tumor growth and facilitate immune escape by suppressing immune responses [9].

Understanding the composition and quantity of immune cells infiltrating tumors is vital for unraveling the complex dynamics of the immune response against cancer. This knowledge not only illuminates the pathways through which the immune system combats or supports cancer but also aids in evaluating the immunogenic impact of anticancer treatments. Such insights are essential for strategically developing combination therapies that can more effectively harness the immune system [5]. Since immunotherapies, including those targeting immune checkpoints, only benefit a select group of patients, analyzing immune cell infiltration in tumor samples before and during treatment offers a promising avenue for discovering new biomarkers [10].

Recent studies have highlighted specific immune cell subsets’ crucial roles in enhancing or dampening the immune response in cancer patients. However, the differential infiltration of certain immune cells of increasing interest across cancers with varied immunogenic potential remains poorly understood. Therefore, this study aims to help fill this gap by examining the density of immune cell groups of interest, including stem-like tumor-infiltrating lymphocytes (TILs), tissue-resident memory (TRM) T-cells, activated-potentially anti-tumor T-cells, early dysfunctional T-cells, late dysfunctional T-cells, and Butyrophilin 3 A (BTN3A) isoforms through assessing their distinct expression across different cancer types by analyzing mRNA co-expression levels of associated cellular biomarkers. Elucidating these key immune cell populations across tumors may help guide the development of tailored treatments and combination therapies taking into account the tumors’ diverse immune milieu.

## 2. Materials and Methods

### 2.1. Patient Cohort and Data Compilation

This investigation utilized a comprehensive dataset merging clinical observations and genetic expression data, nested within the Total Cancer Care^®^ (TCC) Protocol (NCT03977402; Advara IRB# Pro00014441) and the Avatar^®^ project that is conducted within the Oncology Research Information Exchange Network (ORIEN), which is a consortium of 19 cancer centers [11,12]. We utilized clinical and transcriptomic data collected from patients with melanoma (N = 232), bladder urothelial carcinoma (N = 349), ovarian cancer (N = 664) and pancreatic adenocarcinoma (N = 647), representing a spectrum from more responsive to poorly responsive to ICIs. The cohort comprised adult patients with cancer (aged 18 and above), with the study protocol encompassing the collection of tumor, blood, and/or fluid samples as part of standard clinical practice. Data collection extended from each patient’s enrollment in TCC until the data cut-off for this analysis. Written informed consent was obtained from all subjects including the genetic analysis of germline and tumor DNA, alongside the perpetual aggregation of their clinical data. Adhering to the ethical guidelines in the Declaration of Helsinki, the study received approval from all participating entities’ Institutional Review Boards (IRB; Advara IRB# Pro00014441, Tampa, FL, USA).

### 2.2. RNA Sequencing and Data Retrieval

Gene expression relevant to our research interests was interpreted through RNA sequencing, adhering to methodologies elaborated in a previously issued white paper (https://www.asterinsights.com/white-paper/renal-cell-carcinoma-rwd-data/ (accessed on12 February 2024)). The genetic expression data were sourced from the ORIEN database, necessitating the download of numerous FASTQ files for data analysis.

### 2.3. Quantification Analysis of RNA Gene Expression

Gene expression quantification entailed a multistep technical process, initiating with the use of Bbduk software (version 38.96) for trimming adapter sequences from the RNA-Seq reads (https://sourceforge.net/projects/bbmap/ (accessed on 13 August 2023)) [13]. This was followed by aligning the reads to the human reference genome (CRCH38/hg38) using STAR software (version 2.7.3a), available at https://github.com/alxdobin/STAR (accessed on 13 August 2023) [14]. The integrity of the RNA data was verified using the RNA-Seq Quality Control (RNA-SeQC) software (version 2.3.2), https://github.com/getzlab/rnaseqc (accessed on 13 August 2023) [15]. Expression levels were then quantified as Transcripts Per Million (TPM), utilizing the RNA-Seq by Expectation Maximization (RSEM) software (version 1.3.1), https://github.com/deweylab/RSEM (accessed on 14 August 2023) [16], followed by logarithmic transformation with +1 [log_2_(TPM + 1)] and batch effect adjustments via the ComBat function within the sva package (version 3.34.0), https://doi.org/10.18129/B9.bioc.sva (accessed on 14 August 2023) [17].

### 2.4. Generation of Gene Expression Signature Scores

Standardization of gene names was achieved using the NCBI Entrez gene number. Subsequently, the average z-score for each gene expression signature was calculated per the method described by Lee et al. [18].

### 2.5. Enrichment Analysis of Gene Sets

The Gene Set Enrichment Analysis (GSEA) software V.20.3.4., accessible at https://www.gsea-msigdb.org/gsea/login.jsp (accessed on 16 October 2023) [19], was employed alongside hallmark gene sets from the Molecular Signatures Database (MSigDB) V.7.5.1., https://www.gsea-msigdb.org/gsea/msigdb (accessed on 16 October 2023) [19,20,21], to determine significant differences in normalized enrichment score (NES) between responders and non-responders. This analysis involved leveraging curated databases like the Kyoto Encyclopedia of Genes and Genomes (KEGG) and REACTOME for pathway enrichment comparisons. In the end, in an attempt to gain a more comprehensive understanding of the differences in cell types associated with each gene set, the TIMEx “http://timex.moffitt.org” (accessed on 9 March 2023) web portal was used to deconvolute the bulk transcriptomic data of the responder and non-responder samples [22].

### 2.6. Study Outcomes

The primary outcome was the mRNA co-expression levels of specific cellular biomarkers associated with defined types of T-cells, including stem-like TILs (Transcription factor 7 (TCF7), Interleukin-7 receptor (IL7R), CXCR5, CD28, and CD27), TRM T-cells (CD69 and CD103), activated-potentially anti-tumor T-cells (PD1+, CD27+, CD28+, CD137+, Glucocorticoid-induced TNFR-related+ (GITR+)), early dysfunctional T-cells (programmed death-1+ (PD-1+), C-C chemokine receptor type 5+ (CCR5+), TCF7+, T-cell immunoglobulin and mucin-domain containing-3- (Tim-3-)), late dysfunctional T-cells (PD1+, CD38+, CD39+, CD101+, TIM3+), and BTN3A isoforms (BTN3A1, BTN3A2, BTN3A3) assessed across the four cancer types of interest, thereby estimating their differential infiltration.

The secondary outcome was comparing the estimated infiltration of these T-cell types between immunotherapy responders and non-responders. For this analysis, responders were defined as those with at least 2-year survival from the initiation of immunotherapy. Therefore, non-responders were defined by overall survival (OS) < 24 months and responders by an OS ≥ 24 months from the initiation of immunotherapy. Also, we estimated patients’ survival probability based on the estimated infiltration of the T-cell subtypes being studied.

### 2.7. Validation

To validate the predictive value of each significant biomarker, we tested the predictive power of the area under the receiver operating characteristic (AUROC) of each biomarker to predict immunotherapy response in patients with melanoma who are within the OREIN dataset as well as combined data from ten available public datasets, including 672 patients (Table 1) [23,24,25,26,27,28,29,30,31,32,33].

### 2.8. Statistical Analysis

The statistical analysis, conducted by SciPy 1.7.0 software, encompassed the Mann–Whitney U test to compare the median gene expression signature scores of each T-cell type among the four malignancies and the z-score of each T-cell type between immunotherapy responders and non-responders. The Kaplan–Meier analysis and the log-rank test were performed to evaluate the difference in survival probability between patients with low vs. high gene expression of each T-cell type. The AUROC was based on the z-score of immunotherapy responders and non-responders. After correction for multiple testing, which controls the false discovery rate (FDR), two-tailed or FDR-adjusted *p*-values < 0.05 or 0.02, respectively, were established as criteria for significance.

## 3. Results

### 3.1. Patients Characteristics

Table 2 summarizes the demographic and clinical characteristics of 1892 patients analyzed in our study, stratified into specific cancer categories: 232 with cutaneous melanoma, 664 with ovarian cancer, 647 with pancreatic adenocarcinoma, and 349 with bladder cancer. The collective mean ± standard deviation (SD) age was 62 ± 13 years, highlighting a broad age range within the cohort. There was a male predominance in three cancer types (>50%), excluding ovarian cancer. The ethnic composition was primarily non-Hispanic white, exceeding 90% across all cancer categories. Notably, cancer stages at initial diagnosis varied, with Stage III being prevalent among most cancers (about one-third), except for pancreatic adenocarcinoma, where a significant portion (approximately 54.1%) were diagnosed with Stage II. Finally, Eastern Cooperative Oncology Group (ECOG) performance status ranged from 0 (fully active) to 4 (completely disabled), with the majority at 0 or 1, indicating that in patients for whom the ECOG status was known, this trended towards better performance status across the group. Our study highlighted the distribution of histological subtypes within ovarian cancer. A significant portion, 442 cases or 66.5%, were classified as advanced epithelial tumors. The most common subtype encountered was serious cystadenocarcinoma, accounting for 44.1% (293 cases) of the study group. This was followed by adenocarcinoma and serous surface papillary carcinoma, making up 13.4% (89 cases) and 9% (60 cases) of the cases, respectively. Rarer forms, such as clear cell and mucinous adenocarcinomas, were observed in smaller numbers, with the former seen in 3.6% (24 cases) and the latter in 1.5% (10 cases) of the patient cohort. Our study on ovarian cancer histological subtypes observed that the epithelial subtype accounted for most cases, comprising 78% (n = 515). The mixed epithelial and mesenchymal subtype comprised 17% (n = 114) of the cases, while the remaining 5% (n = 35) were classified as Sex-cord stromal, germ cell, or miscellaneous types. Serous carcinoma was the most common sub-classification within the epithelial subtype, accounting for 75% (n = 387) of epithelial cases and 58% of all cases in the total sample. Additional forms of epithelial carcinomas observed in the study included endometrioid (9%, n = 48), clear cell (5%, n = 24), and mucinous (3%, n = 14). The remaining cases were classified as other carcinomas (8%, n = 40).

### 3.2. Differential Immune Infiltrating T-Cells Across Four Cancers

The boxplot in Figure 1 illustrates a broad range of estimated T-cell infiltration across ovarian, pancreatic, bladder, and melanoma cancer types. Notably, ovarian cancer showed the lowest median z-score in all T-cell populations, except for bladder cancer, which was lowest in the expression of BTN3A isoforms. In contrast, melanoma exhibited higher median z-scores among most T-cell populations except for TRM T-cells, which were highest in bladder cancer. Pancreatic cancer had a median z-score that was higher than ovarian and lower than melanoma or bladder cancers. The variability in expression of APA T-cells, early dysfunctional T-cells, late dysfunctional T-cells, and BTN3A-related T-cells, as indicated by the interquartile ranges (IQRs) displayed in the boxplot, was higher in melanoma, while stem-like TILs and TRM T-cells were higher in bladder cancer. On the other hand, the expression variability of all T-cell populations was narrower for ovarian cancer only. Obviously, outliers were present in all T-cell populations of the four malignancies, with several cases showing very high z-scores.

As shown in Figure 1A, the ovarian cancer cohorts demonstrated a significantly lower median expression of stem-like TILs than the bladder cancer or melanoma cohorts (*p*-values = 2 × 10^−8^ or 6.35 × 10^−8^, respectively). Similarly, Figure 1B demonstrates significantly lower expression of TRM T-cell expression levels with ovarian cancer compared to pancreatic, melanoma, and bladder cancers (*p* = 7.852 × 10^−9^, *p* = 2.232 × 10^−8^, and 3.862 × 10^−28^, respectively). A more pronounced significant difference was noted when comparing pancreatic and bladder cancers (*p* = 2.62 × 10^−10^). In Figure 1C, ovarian and pancreatic cancers show significantly lower expression levels of APA T-cells compared to bladder (*p* = 1.862 × 10^−9^ and *p* = 1.382 × 10^−12^, respectively) and melanoma (*p* = 4.672 × 10^−12^ and *p* = 5.82 × 10^−12^, correspondingly). Moreover, the differences in median z-scores in melanoma compared to ovarian, bladder, and pancreatic cancer (*p* = 1.472 × 10^−9^, *p* = 9.762 × 10^−6^, and *p* = 7.822 × 10^−8^, respectively) were significantly higher in the estimated infiltration of early dysfunction T-cells (Figure 1D). Significant differences in the expression of late dysfunction T-cells are highlighted in Figure 1E, including ovarian cancer versus bladder cancer (*p* = 4.722 × 10^−7^), pancreatic cancer (*p* = 5.452 × 10^−11^), and melanoma (*p* = 2.112 × 10^−14^) and also between bladder cancer and melanoma (*p* = 0.000507). Finally, significant differences were seen when comparing melanoma to bladder, pancreatic, and ovarian cancers (*p* = 9.152 × 10^−8^, *p* = 4.682 × 10^−9^, *p* = 1.962 × 10^−5^, respectively) in the expression of BTN3A isoforms.

### 3.3. Comparing Immune T-Cell Subtype Infiltration in Patients with Melanoma Treated with Immunetherapy and Testing Association with Survival Outcomes

We examined the differences in the expression levels of the gene signatures of interest between immunotherapy responders and non-responders in melanoma. Melanoma is known to be an immunogenic tumor with a higher likelihood of response to immunotherapy. This was supported by our results, which demonstrated the highest expression levels of most signatures in melanoma compared to the other malignancies, except for TRM T-cells in bladder cancer. In examining the expression levels of six T-cell signatures of interest between immunotherapy responder and non-responder patients with melanoma (n = 123), strong trends were seen for all, as illustrated in the boxplot analysis in Figure 2. Only TRM T-cells showed a statistically significant difference between responders and non-responders regarding the average z-score (adjusted *p*-value = 0.02).

This figure contains six box plots representing the average gene expression z-score distribution of stem-like tumor-infiltrating lymphocytes (TILs), tissue-resident memory (TRM) T-cells, activated-potentially anti-tumor T-cells, early dysfunctional T-cells, late dysfunctional (dys) T-cells, and Butyrophilin 3 A groups (btn grp) in patients with melanoma treated with immunotherapy. These plots are categorized by responder status: responders (Rs) defined as having OS ≥ 2 years and non-responders (NRs) with OS < 2 years. Adjusted *p*-value < 0.05 is considered significant. Adj P.Val: adjusted *p*-value.

The Kaplan–Meier curves shown in Figure 3 demonstrate an association with improved survival in patients with melanoma whose tumors had high-expression levels of all six signatures tested, where significant differences were seen by the log-rank test in stem-like TILs (*p* = 0.0075), TRM T-cells (*p* = 0.00059), activated-potentially anti-tumor T-cells (*p* = 0.0016), early dysfunction T-cells (*p* = 0.013), late dysfunctions T-cells (*p* = 0.005), and BTN3A (*p* = 0.041).

The figure shows the Kaplan–Meier survival curves, each representing the survival probability as overall survival on the Y-axis over 120 months on the X-axis for melanoma patients treated with immunotherapy. Each plot contrasts the survival probability for patients with high (Red) versus low (Blue) expression of specific T-cell gene signatures. These gene signatures included stem-like tumor-infiltrating lymphocytes (TILs), tissue-resident memory (TRM) T-cells, activated-potentially anti-tumor T-cells, early dysfunctional T-cells, late dysfunctional (dys) T-cells, and Butyrophilin 3 A groups (btn grp) genes. A *p*-value (*p*) < 0.05 was significant.

### 3.4. Immune Cell Infiltration Gene Expression Signature Validation

The TRM T-cell gene signature had the best AUROC as a moderate predictor (AUROC = 0.65) associated with improved survival following immunotherapy in the OREIN dataset. The AUROC of the other GE signatures ranged from 0.55 to 0.59, as shown in Table 3. These signatures had a better predictive performance when examined in the public datasets. Table 3 shows that the six gene expression signatures had similarly moderate predictive values of immunotherapy response in patients with melanoma treated with immunotherapy (AUROC = 0.61–0.64).

## 4. Discussion

Understanding the composition and prevalence of immune cells within tumors is critical for decoding the immune response to cancer and gauging the efficacy of immunotherapy regimens. Gaining such insights is paramount for developing treatments that can effectively activate the host’s immune response [34]. Recent investigations have identified specific subpopulations of infiltrating T-cells that are associated with either enhanced or suppressed immune response in patients with cancer. However, the detailed abundance of these immune cells in tumors with varying immunogenic potentials remains largely unexplored [35,36]. The present study was designed to determine the diverse immune T-cell infiltrates across four cancer types by estimating the density of several immune cell groups, including stem-like TILs, TRM T-cells, activated-potentially anti-tumor T-cells, early and late dysfunctional T-cells, and BTN3A isoforms. This research evaluated the unique expression of these immune cell subsets across ovarian, bladder, and pancreatic cancers and melanoma by analyzing the mRNA co-expression levels of associated cellular biomarkers.

The selection of these specific T-cell subsets and BTN3A isoforms for analysis was rooted in their significant roles within the TME. Stem-like TILs, noted for their proliferation capacity and generation of effector T-cells within the TME, provide insights into the durability and adaptability of immune responses against tumors [37]. TRM T-cells, recognized for their prolonged tissue residency and swift antigenic response, are crucial for local tumor surveillance and correlate with more favorable clinical outcomes [38,39]. Investigating activated-potentially anti-tumor T-cells estimates the host’s direct immune attempts at tumor eradication. Exploring both early and late dysfunctional T-cells offers insights into T-cell exhaustion, a significant obstacle to effective anti-tumor immunity, suggesting pathways for therapeutic intervention to revitalize these cells [40]. Furthermore, BTN3A isoforms, which play a role in T-cell activation for tumor destruction, emphasize the potential for new regulatory mechanisms and therapeutic targets within cancer immunotherapy [41,42]. This precisely chosen array aimed to explain the complex dynamics of T-cell behavior within tumors, varying the ability to respond to immunotherapy, providing a comprehensive perspective on potential immunotherapeutic interventions, and advancing cancer treatment strategies.

The most notable result was that this comprehensive analysis reveals a broad spectrum of estimated T-cell infiltrations among solid tumors, reflecting their unique immune microenvironments and responsiveness to immunotherapy. Melanoma exhibited the highest levels of T-cell gene signature expression among the tumors studied, while ovarian cancer showed the lowest. These observations appear to mirror the distinct immunogenic profiles of these tumors and the complexity of tumor–immune cell interactions within patients diagnosed with these malignancies. Specifically, melanoma’s robust immune response to ICIs, often linked to a higher mutational burden, is associated with T-cell infiltration [33,43,44]. This aligns with the understanding that a high mutational load is associated with improved immunotherapy outcomes due to the enhanced potential for neoantigen presentation and T-cell activation [29]. In contrast, the lower T-cell infiltration and gene signature expression observed in ovarian carcinoma may reflect its lower immunogenic nature or a more suppressive TME, which may present challenges for immunotherapy, supporting the need to develop combinatorial strategies that may enhance immune cell recruitment or counter immunosuppressive elements [45].

T-cell priming, activation, and the overall immune response to tumors depend heavily on the interaction between the tumor antigens and antigen-presenting cells, such as dendritic cells (DCs), along with various co-stimulatory signals within the TME [46]. In our study, we noticed T-cell gene signature expression variability, especially pronounced in bladder cancer and melanoma as indicated by the high IQR in their boxplots, which suggests significant heterogeneity in immune responses within these TMEs, which could result from multiple factors, including the diversity of mutations generating different tumor antigens, the high extent of immune cell infiltration, and the presence of suppressive mechanisms like Tregs or myeloid-derived suppressor cells (MDSCs) [47]. Also, the presence of outliers could indicate individual patients with exceptional immune profiles. At the same time, T-cell gene signature expression variability was more homogenous in ovarian cancer due to the low IQR. This could indicate a more consistent mechanism of tumor antigen presentation, low immune cell recruitment, or a lack of immune diversity in the TME due to tumor/stroma interactions, including a dense fibrotic stroma or vascular characteristics that inhibit immune penetration into the tumor core [48]. This may indicate the importance of cancer-specific approaches in designing immunotherapeutic strategies. Moreover, all examined T-cells showed outliers in the four malignancies, which could hint at a unique patient immune profile that differs from the general trend that is consistent with what is observed in clinical practice. These anomalies may be attributed to distinct genetic, environmental, or treatment-related factors that influence individual responses to immunotherapy, emphasizing the importance of personalized treatment approaches [49].

One obvious finding that emerged from the analysis is that all six predefined T-cell gene sets demonstrated higher expression in immunotherapy responders compared to non-responders, with only TRM T-cells displaying significantly higher expression in responders. This differential gene expression suggests intrinsic immunogenic characteristics of melanoma and a pivotal role of TRM T-cells in mediating effective immune responses in those benefiting from immunotherapy [50,51]. While the higher infiltration by TRM T-cells within the tumors of responders supports their recognized role in sustaining anti-tumor immunity, it raises the possibility that the observed differences may be largely attributed to an overall increase in immune cell presence in responders rather than specific, functionally distinct contributions of the individual T-cell subtypes. Consequently, identifying specific T-cell populations associated with favorable immunotherapy outcomes may lead to more precise patient selection and treatment customization, potentially enhancing therapeutic efficacy; however, it does not provide conclusive evidence that the specific T-cell subtypes we investigated independently drive treatment response [52]. In this study, the Kaplan–Meier survival analysis suggested that patients with melanoma who had high expression levels of all T-cell GE signatures exhibited significantly better survival outcomes than those with lower expression levels. This implies that all T-cell subsets studied constitute interactive elements of a very complex immunogenic TME that ultimately drives the effective anti-tumor response. This observation also supports broadening the investigation of T-cell presence and activity within the TME in an effort to gain valuable insights that could inform the development of biomarkers for predicting immunotherapy outcomes and refining treatment strategies [53].

In the final part of this study, applying the AUROC analysis to evaluate the predictive value of the T-cell gene signatures for immunotherapy response in melanoma revealed that the TRM gene signature emerged as the most robust indicator, displaying superior predictive value. This finding further supports the pivotal role of TRM T-cells in the immune response to immunotherapy in melanoma. This is likely partly due to TRM cells’ capacity for persistent tissue residency and long-term immune surveillance [54]. According to these data, we can infer that the TRM gene expression’s potential as a biomarker for predicting outcomes in melanoma immunotherapy facilitates the identification of patients more likely to derive benefits from such treatments. This finding supports further exploration and validation of these gene signatures in broader patient cohorts and across different cancer types, as well as a deeper investigation into the biological mechanisms underlying these signatures. The goal is to uncover novel targets for amplifying immunotherapy efficacy, thereby advancing the field of personalized cancer treatment strategies. Still, we must point out that the observed variability in AUROC values between the OREIN and public datasets presents a complex picture that warrants further exploration. Specifically, while the TRM gene signature showed a higher AUROC in the OREIN dataset, the other biomarkers exhibited better predictive performance in the public datasets as compared to the ORIEN dataset. These observations likely reflect the specific characteristics of the ORIEN cohort, being a “real-world” dataset that reflects all comers among the patient population that presents to an oncology clinic, including broad disease characteristics, performance status, and patient demographics. By comparison, the public datasets used represent clinical trial participants who have to meet strict criteria with regard to the extent of the cancer, performance status, and other required eligibility criteria. Therefore, clinical trial participants are generally more likely to respond to immunotherapy, while “real-world” patient populations tend to be more heterogeneous and have a poorer prognosis but are also more representative of a real-world patient population. In this regard, our findings continue to suggest that TRM T-cells are more likely to play a more pivotal role as a predictive biomarker since they are also significant in the ORIEN cohort.

This study of the differential infiltration of immune cells in four types of tumors provides significant advancements to the field of oncological research, with multiple notable points. The key strength of this study is its use of a large group of patients, selected from the ORIEN database and verified using public databases, increasing the applicability and significance of its results. The study’s thorough research of four different types of cancer—ovarian, pancreatic, bladder, and melanoma—contributes to a comparative assessment of the immunogenic characteristics of these malignancies. In addition, using modern RNA sequencing and bioinformatics techniques to measure mRNA co-expression levels of certain biological biomarkers linked to various immune T-cell populations is a robust methodological approach. Although this study has notable strengths, it is not devoid of limitations of the retrospective aspect of the data compilation, which, although comprehensive, may involve inherent biases associated with patient selection and data collection procedures. Moreover, we recognize that our data could be interpreted as reflecting a more generalized immune activation rather than a subtype-specific effect. Future studies using more granular methodologies, such as single-cell RNA sequencing or spatial transcriptomics, would be valuable to determine whether these T-cell subtypes possess unique roles within the tumor microenvironment or represent components of a broader immune response and elucidate the underlying factors that influence biomarker performance and stress the importance of validating these biomarkers across multiple, diverse cohorts. We also acknowledge that our analysis is limited by the absence of matched non-cancerous control tissue data, which prevents a direct comparison of immune cell infiltration between tumor and normal tissues. Including such controls would have provided valuable insights into the extent of immune cell infiltration relative to baseline levels. Future research should consider incorporating control tissue comparisons to further elucidate the immune landscape across different malignancies, particularly in cancers like ovarian cancer, which exhibited the lowest median T-cell infiltration in our study. When looking at specific ICIs or regimens, we did not find significant differences, which is likely due to the small sample size. This represents a significant limitation, as the mechanisms of action of these ICIs differ and could influence immune cell infiltration in distinct ways. Future studies should aim to collect detailed treatment information to explore whether certain ICIs confer specific immunological advantages or disadvantages, potentially impacting therapeutic strategies and patient stratification. In addition, the interpretation of these findings must be cautiously approached due to the lack of significant *p*-values for most markers, which could result from various factors, including sample size, patient population variability, or the complex nature of the immune response in melanoma.

## 5. Conclusions

The findings indicated low expression levels of key T-cell markers, mainly in ovarian carcinoma. In contrast, they were significantly higher in melanoma, followed by bladder cancer, and moderate in pancreatic carcinoma. In melanoma patients treated with ICIs, higher densities of stem-like TILs, TRM T-cells, early dysfunctional T-cells, late dysfunctional T-cells, APA T-cells, and BTN3A isoforms were associated with improved survival, supporting a central role in anti-tumor immunity.

## Figures and Tables

**Figure 1 cells-13-01993-f001:**
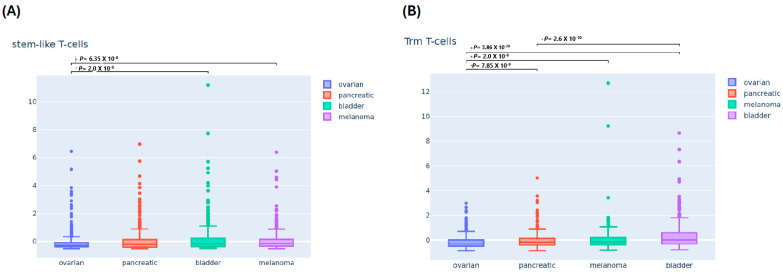
Gene expression of different infiltration T-cells among four malignancies. The boxplots demonstrate the gene expression levels of the signatures corresponding to the T-cell subtypes of interest as well as Butyrophilin 3 A (BTN3A) isoforms among four cancer types. The Y-axis represents gene expression value as a z-score, and the X-axis represents four cancer types: ovarian, bladder, pancreatic, and melanoma. The *p*-value threshold was 0.001. (**A**) Differential expression of stem-like tumor infiltrating lymphocytes (TILs) across four cancer types. (**B**) Expression patterns of tissue-resident memory (TRM) T-cells. (**C**) Activated-potentially anti-tumor T-cells. (**D**) Early dysfunctional T-cell. (**E**) Late dysfunctional T-cell. (**F**) Expression of Butyrophilin 3 A (BTN3A) isoforms.

**Figure 2 cells-13-01993-f002:**
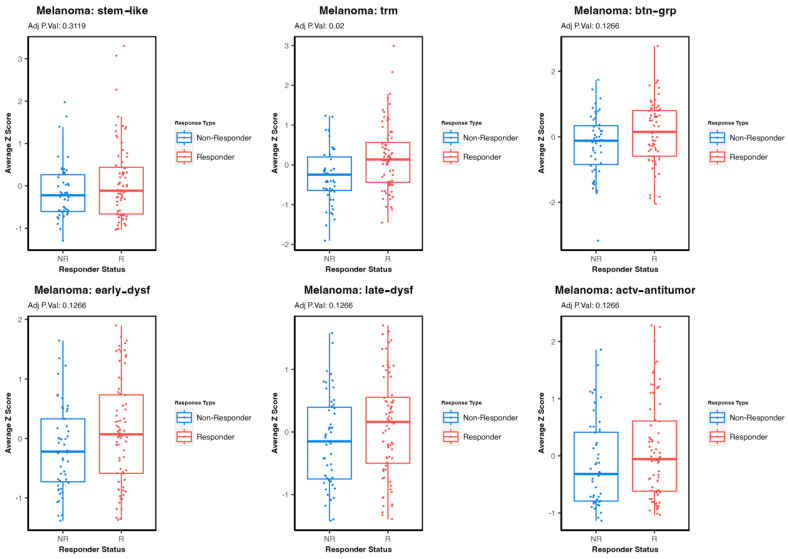
Differential gene expression in immunotherapy responders vs. non-responders in patients with melanoma (n = 123): a box plot analysis.

**Figure 3 cells-13-01993-f003:**
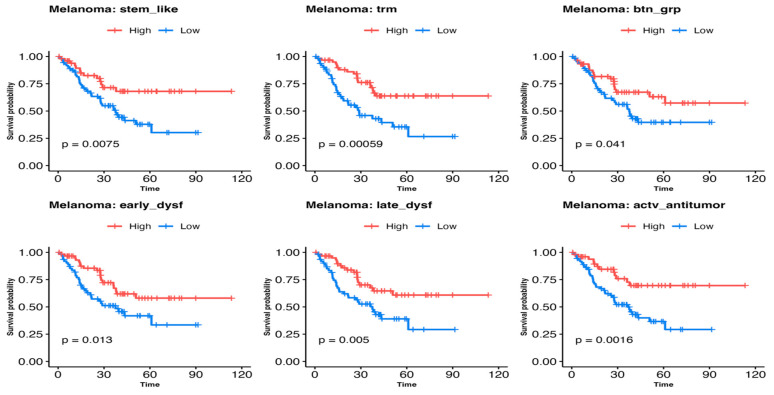
Survival probability of melanoma patients treated with immunotherapy (n = 123): impact of estimated T-cell subtype infiltration.

**Table 1 cells-13-01993-t001:** Public datasets with patients with melanoma on immunotherapy.

Dataset	Cohort	Treatment(s)	Pre /On/Post Treatment	Patients (N)	REF
Du	Du	Ipilimumab/Nivolumab/Pembrolizumab	Pre/On	50	[23]
Gide	Gide_Pre_PD-1_CTLA4	Ipilimumab/Nivolumab/Pembrolizumab	Pre/On	41	[24]
	Gide_Pre_PD-1	Pembrolizumab/Nivolumab	Pre/On	50	
GSE	GSE165278	Ipilimumab	Pre/Post	22	[25]
	GSE158403	Durvalumab	Pre/On	81	[26]
Freeman	Freeman	N/A	Pre/Post	38	[27]
Hugo	HugoLo_IPRES_2016	Pembrolizumab	Pre/On	26	[28]
Lauss	Lauss	Adoptive T-cell therapy	Pre	25	[29]
Lee	Lee	Pembrolizumab/Nivolumab	Pre/On	78	[30]
Liu	Liu	Pembrolizumab/Nivolumab	Pre/On	122	[31]
Riaz	Riaz	Nivolumab	Pre/On	98	[32]
Van Allen	VanAllen_anti-CTLA4_2015	Ipilimumab	Pre	41	[33]

CTLA4: cytotoxic T-lymphocyte antigen-4, N/A: not available, No: number, PD-1: programmed death-1, Ref: references.

**Table 2 cells-13-01993-t002:** Patients’ demographics and characteristics.

Characteristic	Total (N = 1892)	Cutaneous Melanoma (N = 232)	Ovarian Cancer (N = 664)	Pancreatic Adenocarcinoma (N = 647)	Bladder Urothelial Carcinoma (N = 349)
**Age in years** Mean ± SD	62 ± 13	59 ± 14	59 ± 13	63 ± 13	68 ± 11
**Sex, n (%)** Female Male	1141 (60.3) 751 (39.7)	89 (38.4) 143 (61.6)	664 (100) 0 (0)	301 (46.5) 346 (53.5)	87 (24.9) 262 (75.1)
**Ethnicity, n (%)** Hispanic Non-Hispanic Unknown	94 (5.0) 1749 (92.4) 49 (2.6)	9 (3.9) 217 (93.5) 6 (2.6)	39 (5.9) 618 (93.1) 7 (1.1)	32 (4.9) 603 (93.2) 12 (1.9)	14 (4.0) 311 (89.1) 24 (6.9)
**Race, n (%)** African American American Indian or Alaska Native Asian Native Hawaiian or Other Pacific Islander White Other Unknown	55 (2.9) 11 (0.6) 19 (1.0) 2 (0.1) 1757 (92.9) 24 (1.3) 24 (1.3)	1 (0.4) 1 (0.4) 0 (0) 0 (0) 226 (97.4) 1 (0.4) 3 (1.3)	21 (3.2) 8 (1.2) 9 (1.4) 2 (0.3) 610 (91.9) 6 (0.9) 8 (1.2)	23 (3.6) 2 (0.3) 7 (1.1) 0 (0) 601 (92.9) 7 (1.1) 7 (1.1)	10 (2.9) 0 (0) 3 (0.9) 0 (0) 320 (91.7) 10 (2.9) 6 (1.7)
**Cancer stage at initial diagnosis, n (%)** Stage I Stage II Stage III Stage IV Unknown	282 (14.9) 527 (27.9) 526 (27.8) 324 (17.1 193 (10.2)	25 (10.8) 48 (20.7) 73 (31.5) 43 (18.5) 43 (18.5)	97 (14.6) 74 (11.1) 292 (44.0) 129 (19.4) 72 (10.8)	117 (18.1) 350 (54.1) 47 (7.3) 77 (11.9) 36 (5.6)	43 (12.3) 55 (15.8) 114 (32.7) 75 (21.5) 42 (12.0)
**Performance status (ECOG), n (%)** 0 1 2 3 Unknown	386 (20.4) 282 (14.9) 48 (2.5) 8 (0.4) 1168 (61.7)	39 (16.8) 11 (4.7) 4 (1.7) 1 (0.4) 177 (76.3	140 (21.1) 117 (17.6) 22 (3.3) 4 (0.6) 381 (57.4)	141 (21.8) 114 (17.6) 12 (1.9) 1 (0.2) 379 (58.6)	66 (18.9) 40 (11.5) 10 (2.9) 2 (0.6) 231 (66.2)

ECOG, Eastern Cooperative Oncology Group; N, number of patients; SD, standard deviation.

**Table 3 cells-13-01993-t003:** The AUROC of six gene signatures in different datasets of patients with melanoma on immunotherapy.

T-Cells Populations	Stem-like TILs	TRM T-Cells	Activated-Potentially Anti-Tumor T-Cells	Early Dysfunction T-Cells	Late Dysfunction T-Cells	BTN3A Group-Related T-Cells
OREIN dataset	0.554	0.655	0.586	0.593	0.588	0.594
Public datasets *	0.633	0.605	0.633	0.619	0.638	0.625

* Average AUROC. AUROC: area under the receiver operating characteristic, BTN3A: Butyrophilin 3 A groups, OREIN: Oncology Research Information Exchange Network: TILs: tumor-infiltrating lymphocytes, TRM: tissue-resident memory.

## Data Availability

All results relevant to this study are included in the article. Data are available upon reasonable requests to the corresponding author.

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
