# Peer review of "Differential Infiltration of Key Immune T-Cell Populations Across Malignancies Varying by Immunogenic Potential and the Likelihood of Response to Immunotherapy"

_cells, 2024, doi:10.3390/cells13231993_

Round 1
Reviewer 1 Report
Comments and Suggestions for Authors
This paper describes T cell signatures based on RNA sequencing data from databases that defined by gene sets: stem-like TILs (Transcription factor 7 (TCF7), Interleukin-7 receptor (IL7R), CXCR5, CD28, and CD27), TRM T cells (CD69 and CD103), activated-potentially anti-tumor T cells (PD1+, CD27+, CD28+, 170 CD137+, Glucocorticoid-induced TNFR-related+ (GITR+)), early dysfunctional T cells (Programmed death-1+ (PD-1+), C-C chemokine receptor type 5+ (CCR5+), TCF7+, T cell immunoglobulin and mucin-domain containing-3- (Tim-3-)), late dysfunctional T cells 173 (PD1+, CD38+, CD39+, CD101+, TIM3+), and BTN3A isoforms (BTN3A1, BTN3A2, BTN3A3). Here, four different cancer entities are investigated for differential expression of these six gene sets. Authors show association of all six gene sets with responder status and survival.
Major aspects:
1) I am uncertain if the six gene sets are demonstrated to have any heuristic value. They were predefined based on the hypothesis that these six gene sets and associated T cell subtypes behave differently between cancer entities and responder status. However, this was exactly not shown. Instead all six gene sets were upregulated in responders (although only one significant, admittedly).
There was no differential expression of gene sets between groups (responder versus non-responder, the four cancer subtypes). All six gene sets were higher in responder, one of them significant. This is expected.
This does not disqualify the study, but questions if validation – as authors claim – was successfully performed. Wouldn‘t these results be perfectly explained by generally higher infiltration of responding tumors by T cells, with marginal or no contribution of the measured gene sets?
2) I understand that higher AUROC in the OREIN dataset for TRM cells is interpreted as validation for the approach. However, why is this not seen in the public dataset (where the opposite effect is found)?
Author Response
To: Reviewer 1,
Journal of Cells
Special issue: Cellular and Molecular Mechanisms in Immune Regulation
Resubmission Date: November 21st, 2024
Dear Reviewer,
Thank you for reviewing our manuscript “Immune T-cell Populations Infiltration Variability Across Malignancies: Correlations with Immunogenicity and Immuno-therapy Response” ID (cells-3298868). We appreciate the time and effort you have dedicated to providing valuable feedback on our manuscript. We have incorporated changes that reflect your suggestions and highlighted the changes within the manuscript in track changes.
Response to the comments
Comment 1: I am uncertain if the six gene sets are demonstrated to have any heuristic value. They were predefined based on the hypothesis that these six gene sets and associated T cell subtypes behave differently between cancer entities and responder status. However, this was not exactly shown. Instead, all six gene sets were upregulated in responders (although only one was significant, admittedly). There was no differential expression of gene sets between groups (responder versus non-responder, the four cancer subtypes). All six gene sets were higher in respondents, one of them being significant. This is expected. This does not disqualify the study but questions if validation – as authors claim – was successfully performed. Wouldn‘t these results be perfectly explained by generally higher infiltration of responding tumors by T cells, with marginal or no contribution of the measured gene sets?
Response 1: We appreciate the reviewer's thoughtful feedback. The purpose of our selection of these six gene sets was grounded in emerging literature that highlights the relevance of these specific T cell subtypes in immunotherapy responses. However, your point is valid in that our results indicate a general trend of higher T-cell infiltration in responders rather than distinct, subtype-specific expression patterns but also highlight the pivotal role of TRM T-cells in mediating effective immune responses.
To address this, we added a more nuanced discussion in the manuscript to reflect this potential interpretation. We also emphasized that while our findings support a broader association between immune cell presence and therapeutic response, they do not definitively establish subtype-specific mechanisms (lines 401-416). We also clarified the limitations of our study, highlighting the need for further research to disentangle the contribution of these specific T cell subsets from general infiltration patterns (Lines 467-474).
Comment 2: I understand that higher AUROC in the OREIN dataset for TRM cells is interpreted as validation for the approach. However, why is this not seen in the public dataset (where the opposite effect is found)?
Response 2: Thank you for pointing out this observation. The higher AUROC value for TRM cells in the OREIN dataset likely reflects the specific characteristics of the cohort, being a ‘real world” dataset that reflects all comers among the patient population that presents to an oncology clinic, including broad disease characteristics, performance status, patient demographics. By comparison, the public datasets represent clinical trial participants who have to meet strict criteria with regard to the extent of the cancer, performance status and other required eligibility criteria. Therefore, clinical trial participants are generally more likely to respond to immunotherapy, while “real world’ patient populations tend to be more heterogeneous but also more representative of a real-world patient population. In this regard, our findings continue to suggest that TRM is more likely to play a more pivotal role as a predictive biomarker since it is also significant in the ORIEN cohort. We discussed this in the discussion section (lines 438-452).
Sincerely,
Ahmad A. Tarhini, M.D., Ph.D.
Senior Member, Departments of Cutaneous Oncology and Immunology
- Lee Moffitt Cancer Center and Research Institute
Professor, Department of Oncologic Sciences
University of South Florida Morsani College of Medicine
10920 McKinley Dr., Tampa, FL 33612
Tel: 813-745-8581 | Email: Ahmad.Tarhini@moffitt.org
Reviewer 2 Report
Comments and Suggestions for Authors
In this article, Eljilany and colleagues examined the correlation of tumor infiltrating lymphocytes (TILs) and other biomarkers with ICI responsive and non-responsive solid tumor cancer patients from multiple cancer centers. The authors showed that ICI treated melanoma patients have predominant TILs, and they have improved survival as compared to ovarian cancer where the response is lowest. The authors suggested that different combination therapy is needed for various cancer types and analyzing this immune landscape is important to determine the best approach for a particular cancer type and even for an individual patient. The manuscript is very well written. Although the analysis was carefully performed some clarifications are needed to strengthen the rational of the study and the conclusions.
Comments are as follows:
Comments:
1. In case of Figure 1, the authors checked the gene expression of different T-cells in different cancers. Is there control tissue data available from these patients to show even in ovarian cancers how much infiltration is happening as compared to control tissue samples?
2. In Figure 2 and 3 the authors created 2 groups with ICI responders vs. non-responders. Creating a subgroup or table will be helpful to see in both groups what kind of ICI has been applied? Is there any advantage/ disadvantage of CTLA-4 inhibitor over PD-1 or vice-versa?
Author Response
To: Reviewer 2,
Journal of Cells
Special issue: Cellular and Molecular Mechanisms in Immune Regulation
Resubmission Date: November 21st, 2024
Dear Reviewer,
Thank you for reviewing our manuscript “Immune T-cell Populations Infiltration Variability Across Malignancies: Correlations with Immunogenicity and Immuno-therapy Response” ID (cells-3298868). We appreciate the time and effort you have dedicated to providing valuable feedback on our manuscript. We have incorporated changes that reflect your suggestions and highlighted the changes within the manuscript in track changes.
Response to the comments
Comment 1: In the case of Figure 1, the authors checked the gene expression of different T-cells in different cancers. Is there control tissue data available from these patients to show, even in ovarian cancers, how much infiltration is happening compared to control tissue samples?
Response 1: Thank you for highlighting this important point. Unfortunately, control tissue data from matched non-cancerous samples are not available in our study cohort, as the data we used were derived from tumor specimens collected under the Total Cancer Care® (TCC) protocol, which focuses on cancerous tissues. However, we recognize the value of including control tissue comparisons to contextualize the extent of immune cell infiltration and provide a more comprehensive understanding of the tumor immune microenvironment. We included a statement acknowledging the absence of control tissue data in the limitation section of our study (lines 474- 481).
Comment 2: In Figures 2 and 3, the authors created 2 groups with ICI responders vs. non-responders. Creating a subgroup or table will be helpful to see what kind of ICI has been applied in both groups. Is there any advantage/ disadvantage of CTLA-4 inhibitor over PD-1 or vice versa?
Response 2: Thank you for this insightful comment. When looking by specific ICI or regimen we did not find significant differences likely limited by the sample size. This was highlighted in the limitation section (lines 481- 487).
Sincerely,
Ahmad A. Tarhini, M.D., Ph.D.
Senior Member, Departments of Cutaneous Oncology and Immunology
- Lee Moffitt Cancer Center and Research Institute
Professor, Department of Oncologic Sciences
University of South Florida Morsani College of Medicine
10920 McKinley Dr., Tampa, FL 33612
Tel: 813-745-8581 | Email: Ahmad.Tarhini@moffitt.org
Reviewer 3 Report
Comments and Suggestions for Authors
This study is important because there is constant need of characterization of the tumor microenvironment to design better therapeutic strategies which could serve the treatment purpose to the patients in a better way. The mRNA co-expression results showed interesting profiling of the cellular biomarkers which is definitive to specific cell types in the tumor microenvironment in four types of carcinomas. The feasibility and the effectiveness of the available immunotherapy also be seen related to over 24 months of survival and infiltration of those T cell subtypes characterized. Overall, the manuscript is well written covering all the findings. The background and the related references are well documented.
Although there are two things that can be improved before it is considered to be published. The figures and legends are faded, and not prominent to view clearly. Please make the figures and the figures legends readable.
Please elaborate further the importance of prognostic value of the tumor T cell gene expression based on four types of carcinomas described in the manuscript with their corresponding immunotherapy strategies briefly.
Author Response
To: Reviewer 3,
Journal of Cells
Special issue: Cellular and Molecular Mechanisms in Immune Regulation
Resubmission Date: November 21st, 2024
Dear Reviewer,
Thank you for reviewing our manuscript “Immune T-cell Populations Infiltration Variability Across Malignancies: Correlations with Immunogenicity and Immuno-therapy Response” ID (cells-3298868). We appreciate the time and effort you have dedicated to providing valuable feedback on our manuscript. We have incorporated changes that reflect your suggestions and highlighted the changes within the manuscript in track changes.
Response to Reviewer Comments
Comment 1: The figures and legends are faded and not prominent enough to be viewed clearly. Please make the figures and the figures legends readable.
Response 1: Thank you for your comment. We improved the figures' quality
Comment 2: Please elaborate further on the importance of the prognostic value of the tumor T cell gene expression based on four types of carcinomas described in the manuscript with their corresponding immunotherapy strategies briefly.
Response 2: Thank you for this valuable suggestion. We agree that expanding the prognostic value of tumor T cell gene expression and linking it to relevant immunotherapy strategies would strengthen the manuscript. We have already discussed this in our discussion section (lines 364 – 378).
Sincerely,
Ahmad A. Tarhini, M.D., Ph.D.
Senior Member, Departments of Cutaneous Oncology and Immunology
- Lee Moffitt Cancer Center and Research Institute
Professor, Department of Oncologic Sciences
University of South Florida Morsani College of Medicine
10920 McKinley Dr., Tampa, FL 33612
Tel: 813-745-8581 | Email: Ahmad.Tarhini@moffitt.org
Round 2
Reviewer 1 Report
Comments and Suggestions for Authors
My comments have been adequately addressed.
Reviewer 2 Report
Comments and Suggestions for Authors
The authors addressed my concerns. The manuscript can be considered for publication.